# Enhancing Robotic Manipulation: AR-Powered Data Collection for Learning from Demonstration

Thilina Tharanga Malagalage Don[1], Hamid Khayyam[1], and Ehsan Asadi[1]

[1]School of Engineering, RMIT University

*Abstract*—Integrating robotic manipulators into everyday households faces the significant challenge of allowing them to be taught skills in a natural and humanly understandable way. Although learning-from-demonstration (LFD) shows promise, its reliance on quality data and cumbersome demonstration methods limits its broader application. This paper presents a comparison study on the performance of machine learning models, trained using task demonstration carried out via two traditional methods, two traditional methods augmented with augmented reality (AR), and one augmented reality based method. We compare the performance of these input methods against three ML models and two input data modalities. The results demonstrate the advantage of using AR augmented methods in data collection for LFD and the pure AR method nearly matches the performance of the highest performing AR augmented traditional method while having no drawbacks of the traditional methods.

*Index Terms*—AR, LFD, ROS, HRI

## I. INTRODUCTION

Learning from Demonstration (LfD) seeks to capture and replicate the skills or behaviours shown by humans or machines. Although LfD algorithms excel at mastering short-horizon tasks, they require large amounts of data, and their reliance on expert demonstrations which are considered to be high-quality data poses a challenge. Research suggests that the effectiveness of learned policies is directly related to the quality of the input data, based on the underlying assumption of LFD, that demonstrations represent optimal solutions. However, this assumption often falls short in real-world scenarios. Gathering expert demonstrations is inherently resource intensive and requires significant domain expertise and extensive time investment. This is especially true for tasks involving robotic manipulators with high degrees of freedom (DOFs), where human control of the system presents substantial difficulties even for expert demonstrations due to repeated cognitive and physical demands and constrains.

Current demonstration techniques such as teleoperation[16, 8], passive observation [1], and kinesthetic teaching[5] introduce complexities to the demonstration process, which in turn results in sub-optimal demonstrations[17]. The influence of the human teacher's proficiency on the quality of demonstration data was empirically examined in [10]. As task complexity increases, the demand for precise physical guidance escalates, often exceeding the capabilities of novice human teachers and reducing repeated accuracy of expert teachers.

Augmented Reality (AR) has the potential to reduce the cognitive and physical burden in the demonstration process for LFD [19, 20, 14, 7, 11, 15] as it fundamentally allows

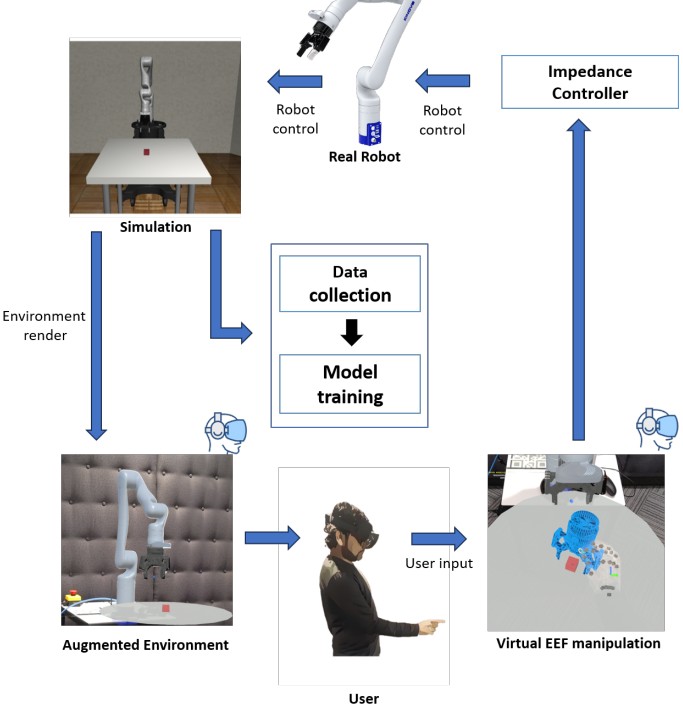

Fig. 1. Architecture: Augmented reality & Real robot blended manipulation teaching environment

to blend the virtual world with the real world seamlessly. AR enables more efficient, accurate, and user-friendly avenues for controlling robots. Specifically, AR has been utilized for reliable visual feedback which reduces the cognitive demand and AR interactive holograms and hologram augmented robots have been used to reduce the physical demand of controlling robots via reducing the forces required and integration of AR supported intuitive high level controllers.

Studies have shown that LFD policy performance can be significantly improved through high quality data which are distinctly characterized by their smother jerk free trajectories and overall and intermediate error free nature. AR based robot arm control methods have shown that they indeed produce trajectories with these qualities thus showing the potential of using AR as an effective tool for generating quality data for LFD policy training demonstrations.

Apart from these, the additional sensing modalities that come with these AR techniques, such as spatial awareness

and egocentric sensing, could become the key enablers that solve long-standing issues with LfD techniques.

While existing literature suggests that AR-based demonstration interfaces offer more intuitive, user-friendly, and less cognitively demanding experiences for human demonstrators, there remains a notable gap in empirical research directly comparing LfD performance trained with data collected through various AR enabled input modalities. This comparison is critical, as the effectiveness of LfD systems often hinges on the inherent characteristics of demonstrations, which can vary significantly between these varies input methods.

Moreover data collection methods both traditional and AR powered must be further combined and optimized to address key challenges, including ensuring clarity and ease of demonstration, accommodating higher DOFs without compromising usability, and simplifying the mapping process between the demonstration and the target task. Addressing these challenges is essential to fully leverage the potential of AR and LFD for facilitating the generation of high-quality demonstrations for effective real world applicable AR, LFD and robotic manipulation.

We introduce an AR assisted data collection platform that allows users to collect task demonstrations through 1) conventional methods (kinesthetic, joystick) via visual observation from a display. 2) AR-enabled holographic visual feedback for conventional methods (AR augmented traditional) and 3) AR and impedance control enabled method (AR-live).The system is supported by a popular robotic policy bench mark simulator, Real robotic arm control middle ware, holographic client generating digital twin of the simulation environment and robot in AR and a high speed communication back-end aligning all above subsystems with precision and low latency.

The main contributions of this paper are;

- Demonstration collection system supported by a manipulation benchmark simulator (robosuite) and robot operating system (ROS) based robotic manipulator control backend that streams a holographic digital twin of the simulation tied to a real robotic arm, to a client's AR headset using low-latency communication protocols.Resulting platform is intuitive to use and ties simulation to the real robotic manipulator facilitating realistic benchmark task related evaluations.
- We conduct a study comparing the performance of machine learning models trained using data collected through two traditional, two AR augmented traditional and one novel AR and impedance control facilitated robot arm control method (AR-live), on simulated benchmark object lifting task with task randomization. The introduced novel method performs similarly to augmented traditional methods and significantly outperforms traditional methods.
- We investigates effects of policy selection in relation to these data using three LFD and batch offline RL models and Low dimentional vs image input data.

## II. RELATED WORK

End-User Robot Programming Using Mixed Reality evaluated the performance of a Mixed Reality (MR)-based interface for robot programming [3]. The study found that participants were able to program a robot arm to perform pick-and-place tasks more quickly, accurately, and easily using the MR interface than with a traditional 2D interface. The study measured the participants' task completion time, strain, naturalness, and usability, MR interface was found to be significantly better than the 2D interface in terms of all the above criteria. The MR interface was also more usable than the 2D interface. However, the study did not address the generalisation challenges or skill acquisitions combined with MR technologies.

Human-robot interaction for robotic manipulator programming in Mixed Reality (ICRA 2020) present an AR-based robot manipulator trajectory generation system[13, 18, 12, 15]. The system allowed users to define start and end points and use key poses for the robot's trajectory, it also includes path scaling, end effector(EEF) obstacle avoidance, and safety zone visualization[2]. The system features a communication infrastructure built to bridge ROS and unity, where ROS move-it-based manipulator control and unity-based holograms were utilized to define user-friendly, effective manipulator action. The study highlights the potential of combining LFD with the AR control interface introduced as potential future work to address the current skills-learning limitation of the proposed system.

ARC-LfD: Using Augmented Reality for Interactive Long-Term Robot Skill Maintenance via Constrained Learning from Demonstration [9] introduces a kinesthetic teaching-based system augmented by AR that allows users to maintain, update, and adapt learned skills through interaction with the key-frames of learned skills using AR. Users can visualize key frames of the skill, edit them in real-time, and define virtual constraints to guide the robot's relearning process. This, in turn, provides an alternate way to define complex temporal relationships within a demonstration to allow performance enhancement of LDF algorithms. However, the use of kinesthetic teaching for skill acquisition imposes demonstration complexities and requirement for expert demonstrators and the requirement for updating key poses for every environmental change makes system less effective in practical robotic manipulation.

The Benefits of Immersive Demonstrations for Teaching Robots [6] explore the potential of virtual reality (VR) to revolutionize Learning-from-Demonstration (LFD) for robots. The paper demonstrates that VR environments offer an intuitive and efficient way for humans to demonstrate complex tasks, leading to smoother and efficient demonstrations. The study also demonstrates these quality demonstrations results in better performing LFD ML models. The paper compares teleoperation-generated data against VR-generated data over 3 manipulation tasks and concludes the VR data were smoother and shorter, which resulted in learned policies that generated smoother efficient trajectories while requiring fewer data.

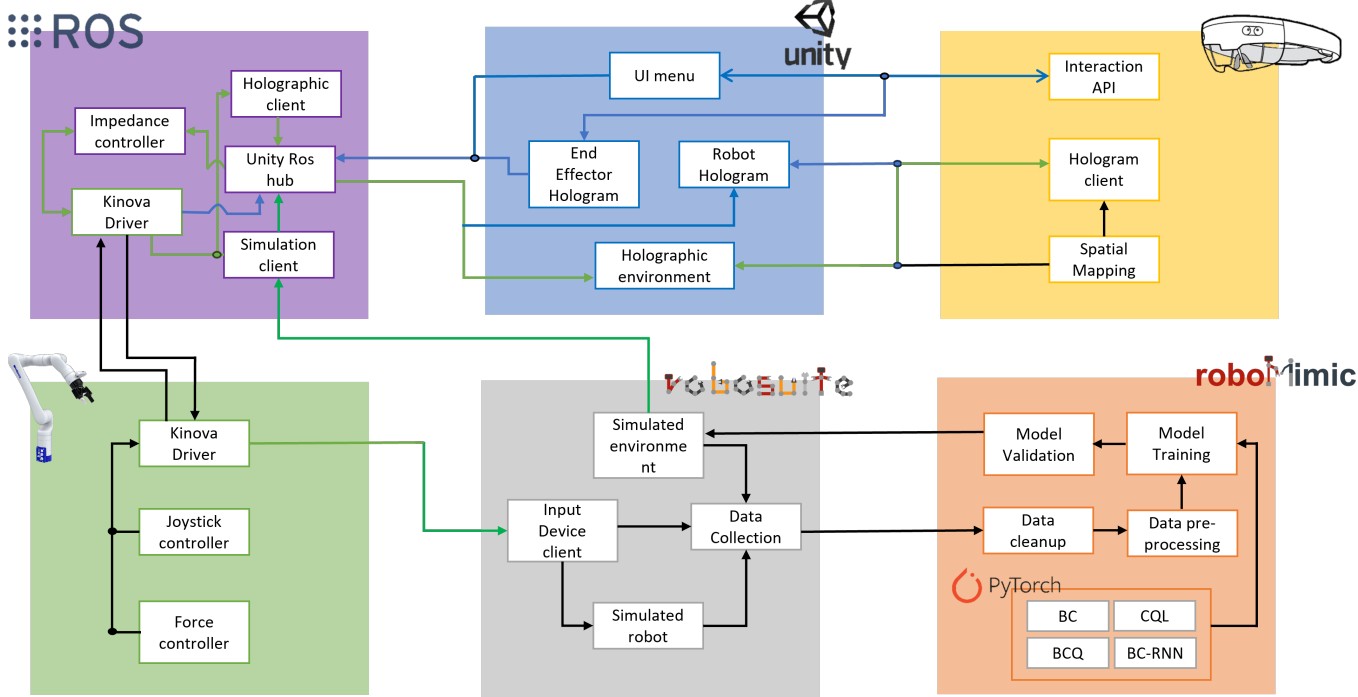

Fig. 2. System implementation: Augmented reality & Real robot blended manipulation teaching environment

However, the study focuses only on final end effector (EEF) trajectories and fail to account for complexities and limitations associated with added DOF and sim to real transfers associated with deployment for real robots.

What Matters in Learning from Offline Human Demonstrations for Robot Manipulation (CoRL 2021) presents a comprehensive investigation into the performance of various learning algorithms in the imitation learning and batch offline learning domain[10, 17]. The study establishes a standardized testing environment and algorithm implementations, paving the way for fair and reproducible comparisons in diverse manipulation tasks. The study compares performance of 6 imitation learning and batch offline RL algorithms over 8 manipulation tasks using demonstration datasets with varying quality. The study concludes the importance of, models with temporal abstractions, quality mistake void data, observation space and the impact of model design on the performance of learned policies through LFD and batch offline RL.

All these factors logically points at using AR based control methods in the demonstration phase for LFD techniques which not only naturally produces quality data without compromises faced by traditional systems.

While several studies have explored the use of AR/VR as a demonstration medium[7, 6], there remains a significant gap in the literature regarding the integration of AR/VR-generated demonstration data with real robotic manipulators and Learning from Demonstration (LFD) algorithms within a standardized robotic manipulation benchmark. This research aims to address this gap by proposing a novel framework that combines these elements, potentially advancing the field of robotic manipulation and human-robot interaction.

## III. TECHNICAL DESIGN: AUGMENTED REALITY & REAL ROBOT BLENDED MANIPULATION TEACHING ENVIRONMENT

Experimental setup shown in Fig. 1. and Fig. 2. mainly encompasses three systems operating in three stages of realism, real world, augmented reality, and pure simulation, integrated on a communication infrastructure built on ROS optimized for precision. The systems can be used in isolation or combined hybrid mode to study the effect of using each system for learning from demonstration on manipulations tasks.

Kinova gen3 6 DOF robot arm equipped with robotiq 2f 85 gripper was used as the main manipulator for the experiment. Majority of the control infrastructure of the manipulator was based on the official ROS drivers provided by Kinova apart from the impedance controller with was built in-house with the use of data from the AR systems processed through a proportional feedback controller with an exponential response curve. Apart from these joystick control and motion through force control(kinesthetic control) was managed by the internal control system of the arm where ROS system kept track of the state of the arm including joint angles, velocity, acceleration, and torque.

HoloLens 2 was used as the untethered self-contained Augmented reality holographic device. Unity 3d, Microsoft Mixed reality toolkit and unity ROS-bridge was used for the application development. The application which was developed based on the Microsoft project [4] architecture using all available features of the device such as spatial computing,

spatial computing precision improvements based on QR codes, spatial mapping, static and dynamic virtual models, hand and eye tracking, voice recognition and natural gesture-based hologram interactions.

MuJoco simulator based Robosuite simulation framework [21] and Robomimic robot learning from demonstration framework [10] was used as the primary data collection environment and validation environment for LFD algorithms. This was due to the environment already being established as a standard bench-marking system for LFD algorithms, input mediums and observation modalities. The environment runs on an OpenAI gym like episodic loop which facilitates convenient data collection and policy validation routines. It also carries the additional advantage of having pre-built data collection sub routines, data conversion functional modules and state of art imitation learning and batch offline algorithm implementations.

Communication between the three systems were designed in the following way. Communication between the ROS pc and the robot arm is handled through standard Kinova supplied move-it interface and hereafter considered as the robot arm interface.

Augmented reality control of arm: Communication takes place between the HoloLens and robot arm interface through unity ROS bridge. Separate ROS service calls were implemented for planning requests, execution requests, planned path demonstrations, robot status updates, obstacle creation, update and deletion, special map exports and homing arm.

Simulated arm control: Controlling of the simulated arm was carried out by designing a completely new input device for the robosuite simulator which maps end effector position of the move-it kinematic solver as the control signal from the input device. This control signal which is compatible with all existing control methods of robosuite then gets interpreted by the selected controller for the simulator instance and moves the arm accordingly.

Hologram updates: At each task two types of virtual object updates are executed in parallel. Updates related to the digital twin of the robot were handled through HoloLens and robot arm interface whereas updates related to the task environment are handled through robosuite and HoloLens. This method has the added advantage of allowing real arm to interact with virtual objects as all three systems, real arm, holographic arm, and simulated arm are exactly mirroring the same action.

## IV. IMPLEMENTATION AND INTERACTION

This part of the paper presents the basic functionality of the system, which is necessary for robot programming and data collection. Fig. 3 gives examples of main virtual objects in our system.

### A. Robot placement and status:

Robot hologram placement was conducted in two stages. Initial placement and digital anchoring from MRTK QR code recognition which also improves spatial tracking precision, and fine tuning through an AR menu with millimeter precision. A

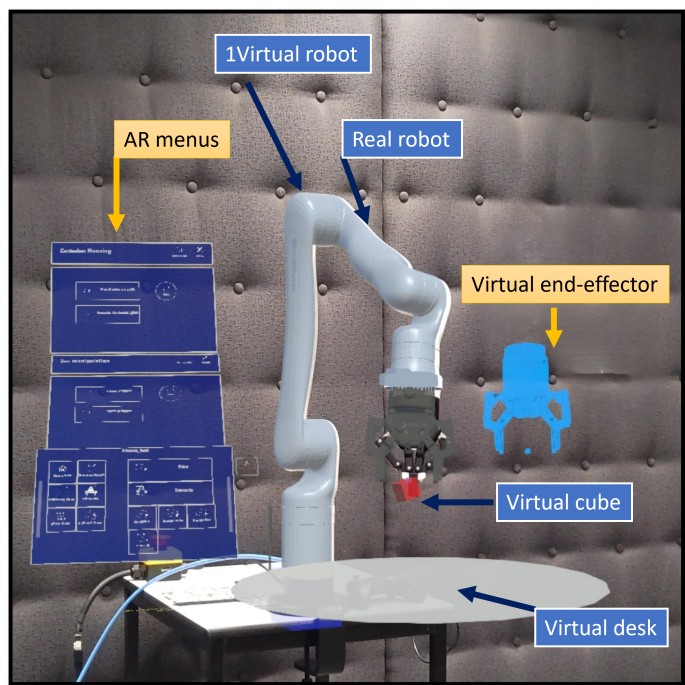

Fig. 3. AR components.

digital twin of the robot arm rendered from official URDF was updated to mirror the robot status (joint angles) at every update cycle to mirror current robot status which was also used to preview intended motion plans.

### B. Trajectory Execution:

Following robot placement finalization, users manipulate a holographic end effector to define the desired end effector path. This end effector mimics the real end effector (EEF) of the robot, allowing for free movement in all six degrees of freedom. while moving this virtual EEF, the impedance controller generates a trajectory for the robot arms EEF to closely follow the virtual EEF, The impedance controller generates the output signal on an exponential curve to achieve both responsiveness and precision. The gripper open/close and initial pose reset options for both virtual and real EEF was provided through AR menus.

### C. Task environment:

For benchmarking models and input modalities, a digital twin of a standard Robosuite manipulation task ("Lift") was constructed in Augmented Reality (AR). A dedicated rendering pipeline ensures in-sync updates between the AR environment and the Robosuite simulation.

### D. Dataset/Training subroutines:

Dataset creation was handles by a modified version of the standard dataset creation module of robosuite which allowed for the new input device and included triggers and reset events for execution event from AR interface. Due to the design of the input method, standard training modules were

compatible with the new datasets created by the proposed methods thus allowing recreation of all available experiments from the standard Robomimic environment for the selected task.

## V. METHODOLOGY

The aim of our study was to develop and evaluate performance of LFD policies trained with data from AR based demonstration methods, AR augmented conventional demonstration methods and conventional demonstration methods against a standard robotic benchmark task. The developed experimental setup (Fig1) was used to collect demonstrations, pre-process and used for training with imitation learning algorithms and offline reinforcement learning in the following manner.

### A. Data Collection

Data collection involves four steps as illustrated in Fig.4. At each simulation instances the position of end effector, cube position and z axis rotation were randomized. Data collection was triggered at each new instance of a simulation. The end of the simulation instance was triggered by either manual reset triggered by user or a task succession. After each end of instance, if the task was successful, the recorded data instance was amended to a hdf5 file, the real robot arm automatically send to the reset pose and the new object instance was rendered relative to the real robot end effector pose. If the reset was triggered by the user, the collected data instance was simply discarded, and a similar reset routine was carried out.

### B. Preprocessing

Each data instance contains a complete simulator snapshot allowing exact conditions to be replayed, different views and data streams to be rendered for different experiments such as low dimensional data, image data, segmented image data etc. Each dataset needs to be pre-processed in two stages to be compatible with LFD algorithms. First each dataset was converted to be compatible with Robomimic framework[10] where metadata related to the task environment were added to the dataset. Finally, another preprocessing step was carried out to render required data from the simulated snapshots to be fed into the LFD algorithm according to the learning run executed

### C. Problem formulation and Model design

The manipulation task was formulated as an infinite horizon discrete time Markov Decision process (MDP)

$$M = (S, A, T, R) \tag{1}$$

where 4-tuple represents state space, action space, transition function, and reward function respectively. The goal was to learn a policy $\pi$ that enables to take an optimum action at each state $s_t$,

$$a_t = \pi(s_t) \tag{2}$$

An offline dataset of trajectories

$$D = \{(s_0^i, a_0^i, r_0^i, s_1^i, \ldots, s_t^i)\}_{i=1}^{N} \tag{3}$$

where N is the number of demonstrations and t is the horizon of the demonstration (during training and policy rollouts $t_{max}$ was 400 episodes), was provided for the models to learn policy $\pi$. Each learned policy was tested periodically in the simulation environment for task success rate. A binary reward function,

$$R(s, a, s') = I(s' \in G) \tag{4}$$

where G is task completion states, was provided for batch offline reinforcement algorithms.

### D. Training

Training was carried out using the Robomimic simulation environments standard training pipeline. BC-RNN algorithm was used for optimizing and comparing the input methods. For initial comparisons following low dimensional data was provided as the input for the model

- Cube position (cartesian) and orientation (quaternion)
- Cube position relative to robot end effector (cartesian)
- Robot proprioception data (six joint angles) and end effector status

For image-based training runs, instead of ground truth cube pose data from the simulator, images from the 3rd person view and robot end effector view were used for training BC-RNN with a ResNet-18 based network estimating the cube pose.

## VI. EXPERIMENTS AND RESULTS

### A. Effectiveness of proposed methods

Fifty demonstrations from each of the traditional, AR assisted traditional and AR + motion planning methods were collected for analysis. All datasets were trained using BC-RNN model with same hyperparameters.

### B. Impact of model selection on task success rate

Datasets with hundred demonstrations each were created using traditional, AR assisted traditional and AR + motion planning methods. Following models were used to train each of the above datasets with same hyperparameters for training runs from each model.

- Behavioral Cloning (BC)
- Behavioral Cloning with recurrent neural network (RNN)
- Batch-Constrained Q-Learning (BCQ)

### C. Model performance with observation space.

Previous training runs were carried out using low dimensional data where the proprioception data, gripper pose, cube pose data obtained from the simulator and 6DOF input data were used for training. Here instead of cube pose data from simulator, images from the 3rd person view and robot end effector view were used for training BC-RNN with a ResNet-18 based network estimating the cube pose.

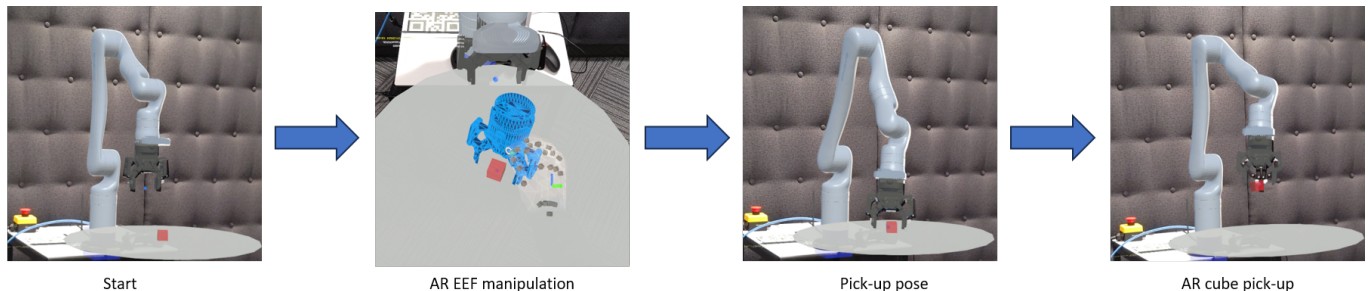

Start | AR EEF manipulation | Pick-up pose | AR cube pick-up

Fig. 4. Data collection flow for AR + impedance control input modality.

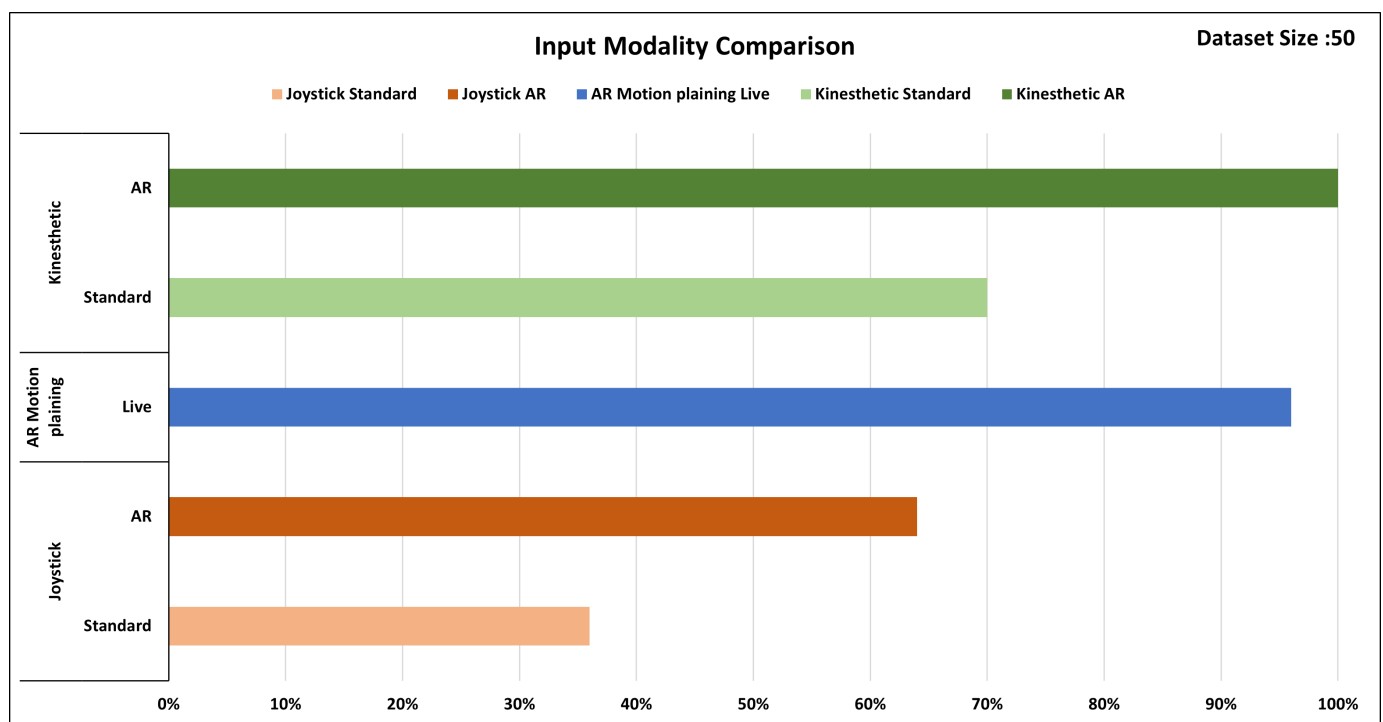

Fig. 5. Task success percentage per input modality.

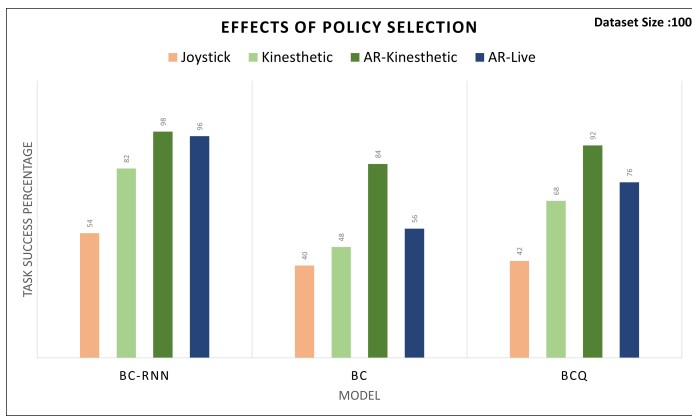

Fig. 6. Effects of policy selection against input modalities(expert user).

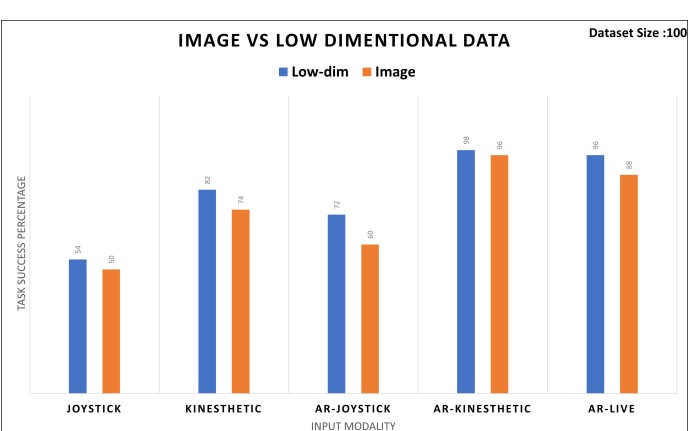

Fig. 7. BC-RNN model performance against low-dimensional data vs Image data.

## VII. DISCUSSION

### A. Effectiveness of the input data modalities

The results Figure 5 and Figure 6 shows that ,Highest performing policies at inference are trained from demonstrations collected from traditional methods augmented with AR, where kinesthetic teaching augmented with AR showing 100% task success-rate.Apart from this the proposed AR + impedance controller method(AR-live) nearly matches the effectiveness of the kinesthetic teaching augmented with AR method while having further advantages of uniformity across different manipulators, ability to operate with simulated manipulators, demonstration via remote teleoperation, improved safety and reduced physical strain.

### B. Policy selection

The results (fig 6) shows that

- All methods shows improved task success rate with the addition of 100% more data where kinesthetic teaching augmented with AR being the only exception due to task success rate already achieving maximum in the first instance.
- On average across BC,BC-RNN,BCQ, Kinesthetic augmented wih AR was the most successful method.
- BC-RNN on average has the highest task success rate across all the input methods showing the importance of models with temporal abstractions in learning manipulation tasks.This results also agrees with the finding of[10]

### C. Observation space

The results (fig 7) shows that the initial experimental results using low dimensional data holds true with image base experiments which shows possibility of sim to real transfer of learned models with minimum modifications.

### ACKNOWLEDGEMENTS

I would like to acknowledge the support of the Intelligent and Adaptive Cobotics Lab and Virtual Experiences Laboratory / Hub for Digital Innovation at RMIT University for this study.

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
