# OpenReview forum: "Enhancing Robotic Manipulation: AR-Powered Data Collection for Learning from Demonstration"
_humanrobotinteraction.org/HRI/2025/Workshop/VAM — HRI 2025 Workshop VAM Submission_

### Official Review · Reviewer_vyxs · 2025-02-21

**Rating:** 6
**Confidence:** 5

**Review:**

Summary:
This paper compares the performance of models trained from demonstrations given by expert users across four input modalities (joystick, kinesthetic, AR kinesthetic, and AR-Live). Their proposed learning from demonstration approach, AR-Live, leverages AR and impedance control. They found that AR-Live performs similarly to AR kinesthetic teaching. They also conclude that a Behavioral Cloning Recurrent Neural Network policy is best across all four input methods compared to Behavioral Cloning and Batch-Constrained Q-Learning.

Strengths:
* Their demonstration collection system is very clearly described and can be easily replicated
* Compared the performance of models trained from demonstrations given across four input modalities (joystick, kinesthetic, AR kinesthetic, and AR-Live), which seems to be a novel contribution
* The close performance between the low dimensional and image data suggests that this approach may transfer well from sim to real

Areas for Improvement:
* There are many typos throughout the paper. These include: missing spaces, capitalization typos, and some grammatical errors. Other specific typos include an extra “at” right before algorithm 2 in section 3C. There are also formatting typos when talking about the algorithms: the “t” in “st” should be a subscript and the entire word “max” in “tmax” should be a subscript. Typo on the word Standard in Figure 5 and dimensional in Figure 7.
* In Section 7B there’s a small typo in “addition of 50% more data”, it seemed to double from 50 to 100 samples which would be a 100% increase. This made me wonder, why do these analyses have different numbers of samples?
* I’d be interested to see more details regarding how the samples were collected. It seems that a total of 400 samples were collected (100 for each input modality). Were these all collected by the same expert user or were participants recruited? How did you account for variations such as differences in the experts’ experience with each input method? Were there any potential learning effects if the expert(s) recorded demonstrations for more than one input category? Were there any training procedures in place? It was mentioned that failed demonstrations were discarded, were there differences in the number of failed demonstrations across each input method?

Other comments:
* In future work, it would be interesting to run a follow up user study to compare some of the other trade-offs mentioned in the paper, such as workload and usability, across these input modalities.
* It would also be nice to follow up on the sim to real transfer, especially across a variety of tasks with different complexities.


Recommendation:
This work is interesting, and seems like a great fit for VAM-HRI. I recommend this paper for acceptance at VAM-HRI this year, but I strongly recommend for the authors to take another pass through their paper to correct typos for the camera ready submission.

---

### Decision · Program_Chairs · 2025-02-26

Accept